# A2X: An Agent and Environment Interaction Benchmark for Multimodal Human Trajectory Prediction

## Abstract

Recent trends in human trajectory prediction are the development of generative models which generate distributions of trajectories. However existing metrics are suited only for single (unimodal) trajectory instances. Furthermore, existing datasets are largely limited to small-scale interactions between people, with little to no agent-to-agent environment interaction. To address these challenges, we propose a dataset that compensates for the lack of agent-to-environment interaction in existing datasets with a new simulated dataset and metrics to convey model performance with more reliability and nuance. A subset of these metrics are novel *multiverse metrics*, which are better-suited for multimodal models than existing metrics but are still applicable to unimodal models. Our results showcase the benefits of the augmented dataset and metrics. The dataset is available at: https://mubbasir.github.io/HTP-benchmark/.

## 1 Introduction

The study of human navigation has long been of interest to various research communities such as computer graphics [10], computer vision [1], cognitive science [33], and robotics [5]. Advancements in these areas have seen widespread practical application in pandemic response, architectural design, urban planning, transportation engineering, crowd management, socially compliant robot navigation, and entertainment. Accordingly, the influence of human navigation research has reached countless individuals and will continue to do so in the foreseeable future.

Most applications rely on simulation models [20], which are sufficiently accurate to human behavior and generalizable to unforeseen circumstances. However, the past five years of predictive modeling in computer vision has achieved significantly better accuracy [23], giving it a strong potential to overtake the longstanding models from computer graphics. This is largely due to the transition from using unimodal, discriminative models [1] that predict a single future trajectory to using multimodal, generative models [7, 24, 18] that predict a distribution of future trajectories, which captures the inherent uncertainty in human decision-making [25, 4]. Despite the evolution of models, however, the accuracy metrics that were introduced with the first unimodal models are still in use today. In order to adapt these fundamentally unimodal metrics to multimodal models, the metrics are computed between each predicted trajectory and the ground truth trajectory, and the minimum error for each metric is reported. This results in a gross overestimation of accuracy that we later show is not consistent with the expected accuracy, which may misguide future research efforts. Furthermore, the minimum value is not actionable, because while it is evident that a state-of-the-art (SOTA) multimodal model can find *an* accurate trajectory, it cannot determine *which* trajectory that is on unseen data. We measure this uncertainty through a decidability metric.

Generalizability cannot be maximized by solely improving upon accuracy metrics. An inaccurate model can be robust by producing realistic trajectories, and an accurate model can fail to be practicable

Submitted to the 35th Conference on Neural Information Processing Systems (NeurIPS 2021) Track on Datasets and Benchmarks. Do not distribute.

by being undecidable. Models can exist on the continuum between these two extremes, making it critical to consider realism and decidability metrics as well.

Furthermore, there is a stark class imbalance in existing datasets. While datasets are abundant in instances where humans are interacting with each other in open spaces [16, 22, 2, 34, 3, 14], they are significantly lacking in both environment information and instances where humans are interacting with their environment. Ultimately, this hinders generalization at a global level and has led to some models being developed without considering environments at all [1, 7].

In this work, we provide an augmented human trajectory prediction dataset that compensates for the lack of agent-to-environment interaction in existing datasets with a new simulated dataset. To understand model performance on this new dataset with more reliability and nuance, we propose a comprehensive set of accuracy, realism, and decidability metrics. A subset of these metrics are novel *multiverse metrics*, which are better-suited for multimodal models than existing metrics but are still applicable to unimodal models. The evaluation using these metrics decisively evidences that the new dataset facilitates better robustness and generalization, that current metrics can be misleading, and that there are still remaining challenges to modeling human trajectories. We finally showcase that realism metrics can also be used to decide which prediction to take from an undecidable multimodal model through the process of *Multimodal Model Collapse*. Henceforth, we refer to humans as agents, since our conceptual framework is broadly applicable, e.g. to robotic and vehicular agents.

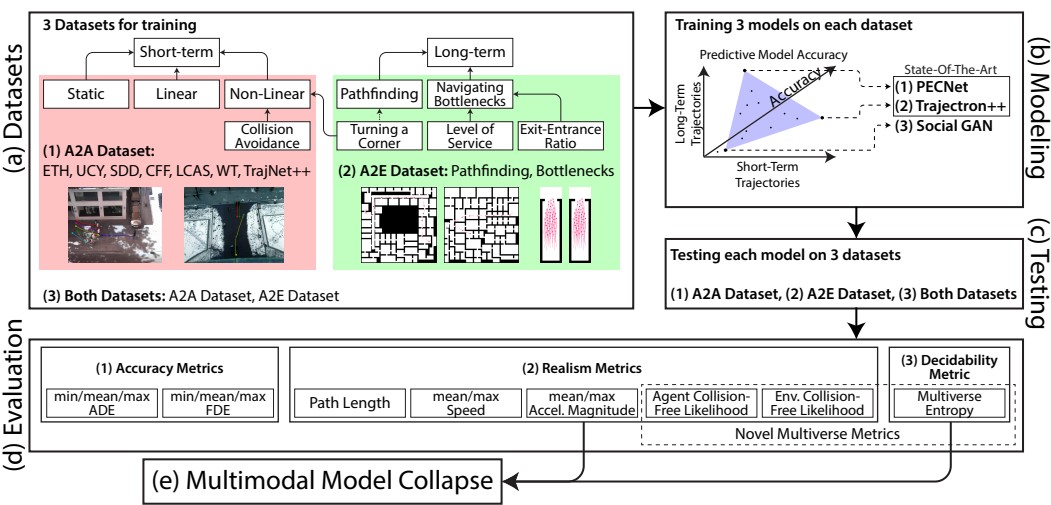

Figure 1: The above framework image shows (a) the differences between the trajectories of existing datasets (A2A) and the novel dataset (A2E), (b-c) the models trained and tested on combinations of A2A and A2E, (d) the proposed set of metrics for evaluating the accuracy, realism, and decidability of models, and (e) a greedy method for selecting the prediction most realistic movement.

## 2   Background and Preliminaries

**Models for Human Trajectory Prediction.** Earlier methods such as Social LSTM [1] and Social Attention [31] proposed a deterministic model which predict a future trajectory given observed trajectories. However, forecasting trajectories inherently introduce the uncertainty in the future, hence the utility of those uni-modal models which predict only one future trajectory is limited. Recent studies [7, 18, 24, 36, 12, 17] assume the multi-modalities in the future human behavior and predict its distribution to embody the uncertainty. In this paper, we focus on three SOTA methodologies to showcase our benchmark dataset: SocialGAN [7], PECNet [18], and Trajectron++ [24].

SocialGAN [7] adopts GAN [6] framework to forecast possible future trajectories and it can avoid collisions among pedestrians by introducing a pooling mechanism that captures between-human interaction. PECNet [18] solves the trajectory prediction problem by first modeling the future goal position distribution using a Variational Autoencoder (VAE) [13], and then predict the future positions by interpolating the observed positions and the estimated goal position. Trajectron++ [24] proposes

a graph structured recurrent model based on conditional VAE [28] to predict the future trajectories. Further details can be found in the Supplementary Materials.

We investigate these three models as the representatives of the various SOTA works. We choose them because PECNet [18] shows an outstanding performance on the long-term trajectory while the short-term trajectory is most well predicted in Trajectron++ [24]. We expect SocialGAN [7], as one of the earliest and most frequently referred models, to be a bound around existing models with respect to PECNet and Trajectron++. Fig. 1.b shows the coverage comparison of SOTA models in terms of the short- and long-term human trajectory prediction accuracy. We differentiate between predictive models of short-term and long-term trajectories on the basis of goal conditioning. A model that is not goal-conditioned will inherently increase in error as the predicted path length increases, sometimes at an exponential rate [24], whereas goal-conditioned models are expected to predict long paths without the same trade-off between path length and error.

**Datasets for Human Trajectory Prediction.** The computer vision and graphics community have collected several human pedestrian trajectory datasets. ETH [21] and UCY [16] are commonly used datasets that contain five outdoor scenes with jointly more than 1,600 pedestrian trajectories. Stanford Drone Dataset (SDD) [22] consists of eight outdoor scenes tracking 19,000 targets including pedestrians, bicyclists, skateboarders, cars, and buses collected from a drone. Stanford Crowd Dataset (CFF) [2] consists of pedestrian trajectories collected within a train station building of size 25m $\times$ 100m for 12 $\times$ 2 hours captured by a distributed camera network. L-CAS 3D Point Cloud People Dataset (LCAS) [34] consists of 28,002 scan frames collected within a university building by a 3D LiDAR sensor mounted on a robot that is either stationary or moving. WILDTRACK (WT) [3] is a collection of annotated dense pedestrian groups captured by seven static HD cameras in a public square for about 60 minutes. The Supplementary Materials provide more details of these datasets. Some datasets, such as TrajNet++ [14], augment upon existing datasets. TrajNet++ combines ETH/UCY, CFF, LCAS, and Wildtrack datasets, as well as a synthetic dataset generated by ORCA [30].

Existing human trajectory datasets have limitations in the sense of embodying interactions. They either do not contain agent-to-environment (A2E) interactions [3], or exhibit limited agent-to-agent (A2A) interactions at small scale in simple environments. We speculate that many self-centered pedestrians are prone to avoid or mitigate, consciously or unconsciously, the influence of the environments and other pedestrians during their navigation. In this work, we are proposing datasets that augment A2E and A2A interactions, which may bring benefits for enhancing learning models by encoding more complex trajectory dynamics.

**Benchmarks for Human Trajectory Prediction.** In computer graphics community [27], trajectories are, in general, measured by motion statistics such as the number of collisions, average speed, average acceleration, and total distance traveled. On the other hand, in machine learning community [14, 1, 7], the most commonly used evaluation metrics for trajectory forecasting models are Average Displacement Error (ADE) and Final Displacement Error (FDE). ADE is the average $L_2$ distance between the ground truth and the predicted trajectories across all future steps. FDE is the $L_2$ distance between the ground truth final destination and the predicted final destination at the end of the future steps. More evaluation metrics in machine learning community are discussed in Supplementary Materials.

ADE and FDE are applicable to unimodal methods which predict only one future sequence that can be compared with the ground truth future sequence. However, as aforementioned in this section, many multimodal trajectory forecasting models assuming uncertainty and multimodality in pedestrians' future behaviors predict $k$ future sequences (usually $k = 20$). Most of these models report the minimum ADE / FDE results among all $k$ predictions, which, in our view, is over optimistic. Not only is this a significant underestimation of the error, but it is also an impossible standard in that these models are incapable of choosing the prediction with the minimum error. In Section 4 of this work, we propose new metrics that can tackle this issue.

# 3  Agent-to-Agent and Agent-to-Environment Interaction Dataset

We propose a comprehensive trajectory prediction dataset **A2X** that consists of a representative set of trajectories, which will enable better generalization under realistic circumstances that are either complex or unsafe and out-of-distribution (OOD) with respect to current datasets.

In order to understand what the shortcomings of current datasets are (Sec. 2), we first taxonomize the characteristics of human trajectories. The TrajNet++ benchmark [14] proposed an initial taxonomy that only considers short-term characteristics, e.g., standing still, moving linearly, or avoiding collisions (Fig. 1.a). While the original taxonomy is sufficient for describing the trajectories in many real datasets and their agent-to-agent (A2A) interactions, models that learn exclusively from these types are insufficient for most applications, which consider environments that have non-navigable regions and time frames longer than 5 seconds, which is the practical limit for most models before they become exponentially erroneous [24]. We have improved upon this by considering long-term characteristics (Fig. 1.a), i.e., pathfinding alone and navigating through crowded bottlenecks. These types of trajectories emerge from agent-to-environment (A2E) interactions, which unfold over a longer time frame than A2A interactions and are essential for navigation within any environment [29].

## 3.1 Agent-to-Agent Interactions

For representing A2A interactions, we make use of each prior dataset described in Section 2: ETH [16], UCY [16], SDD [22], CFF [2], LCAS [34], WT [3], and TrajNet++ [14]. These datasets feature transient interactions between agents and little interaction with the environment, which is made difficult to measure by the frequent unavailability of environment information. Therefore, we approximate environment information based on the principle of stigmergy [19, 11], which observes the self-organization of human navigation along trails. For each position that agents have traveled through in either the training or testing sets of the ground truth, a 1-meter radius around the position is considered to be navigable. This guarantees that predictions with less than 1 meter of displacement from the ground truth at all times will never intersect with the environment. In addition, in order to compensate for the imbalance between A2A and A2E interactions in prior datasets, we propose the generation of synthetic data in addition to that of TrajNet++. While real datasets are valuable for their veridicality, there are logistical limitations that prevent the acquisition of real data in OOD scenarios that are unsafe for human participants or prohibitively expensive from an organizational standpoint.

## 3.2 Agent-to-Environment Interactions

Two such scenarios are used to sample trajectories exhibiting A2E interactions: (1) pathfinding alone in a large, complex environment, which has prohibitive logistical cost and (2) navigating through bottlenecks of varied width with a dense crowd, which can be unsafe. Though simulation models are normally less accurate than predictive models in predicting human trajectories [1], the prevalent Social Force model [10] currently outperforms predictive models in terms of robustness, has been used in several application domains [5, 32, 35], and has ecologically validity in these A2E scenarios, which have not had sufficient real data for training predictive models until **A2X**.

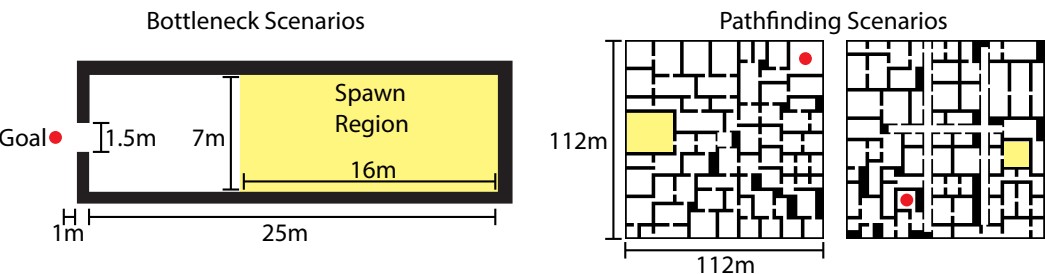

Figure 2: The above images show the exact dimensions of environments from the bottleneck and pathfinding scenarios in A2E.

We leverage the Social Force model to simulate 236 scenarios of a single agent navigating between random points in complex $112 \times 112$ m$^2$ environments from [29] (Fig. 2). This produces long-term isolated interactions between single agents and the environment. We then use the same model to simulate well-studied bottleneck scenarios [26, 9] in a $25 \times 7$ m$^2$ room that vary in terms of (a) the density of agents (Level of Service) from $\{0.2, 0.4, 0.6, 0.8, 1.0\}$ agents/m$^2$ and (b) the ratio between the width of the bottleneck and the width of the room (Exit-Entrance Ratio) from $\{0.2, 0.3, 0.4, 0.6, 0.7\}$ (Fig. 2). A total of 398 scenarios have been generated across all combinations of Level of Service and Exit-Entrance Ratio. This produces long-term interactions between agents as a result of the constricting environment. Exact environment information has been provided for both

types of scenarios. We later show that current models trained on existing A2A datasets are unable to generalize to these critical scenarios, but with the addition of training data on these scenarios, the accuracy of predictions significantely improves.

# 4 Accuracy, Realism, and Decidability of Human Trajectory Prediction

We propose a total of 15 accuracy, realism, and decidability metrics (Fig. 1.d). These metrics are either borrowed from computer vision and computer graphics literature [21, 1, 27, 8] or newly developed *multiverse metrics*, which assess the A2A and A2E interactions of both multimodal models with $k > 1$ and unimodal models with $k = 1$.

## 4.1 Preliminaries

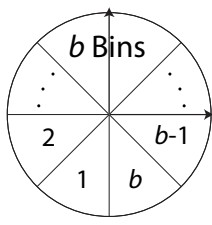

In accordance with both unimodal and multimodal predictive models, we utilize the following notation for their predictions. A prediction scenario is defined by a set of $n$ agents present in an environment $\mathbf{E}$ at the same time. Each agent $a$ has $t_p$ frames of past position data as input and $t_f$ frames of future position data for ground truth $\mathbf{Y}_{a,0} \in \mathbb{R}^{t_f \times 2}$ and for each prediction $\widehat{\mathbf{Y}}_{a,j} \in \mathbb{R}^{t_f \times 2}$, where $0 \leq j < k$. All position data is in meters and has a frame rate of $1/\Delta t$ hertz based on the dataset. The position at the $t$-th frame is $\mathbf{Y}_{a,0,t} \in \mathbb{R}^2$ for the ground truth and $\widehat{\mathbf{Y}}_{a,j,t} \in \mathbb{R}^2$ for prediction $j$, where $0 \leq t < t_f$. We then compute the velocities corresponding to the ground truth $\mathbf{V}_{a,0} \in \mathbb{R}^{(t_f-1) \times 2}$ and each prediction $\widehat{\mathbf{V}}_{a,j} \in \mathbb{R}^{(t_f-1) \times 2}$.

Figure 3: This images shows how $b = 8$ bins would be arranged in 2D space.

Many of the following metrics make use of aggregate functions. For any $d$-dimensional vector $\mathbf{v} \in \mathbb{R}^d$, we denote the minimum value by $\Omega(\mathbf{v})$, the mean value by $\Theta(\mathbf{v})$, and the maximum value by $\mathrm{O}(\mathbf{v})$. For a matrix of $d$-many 2D vectors $\mathbf{V} \in \mathbb{R}^{d \times 2}$, function $\Xi(\mathbf{V}, b)$ transforms the 2D vectors into a probability distribution $\mathbf{p} \in \mathbb{R}^b$ over a vector of $b$-many equiangular bins, which radiate from the origin (Fig. 3). Finally, we denote the $L_2$ norm by $\| \cdot \|$.

## 4.2 Accuracy Metrics: Comparison to Ground Truth

Accuracy metrics from computer vision literature are responsible for comparing the ground truth with the predictions based on the displacement error.

**Average Displacement Error (ADE).** ADE is computed for each prediction $j$ as $\mathbf{a}_j$, the average distance between a position in the ground truth and a position in the prediction across $t_f$ frames (Eq. 1) [21]. It is then aggregated across the $k$ predictions in three ways: minimum, mean, and maximum, which offers a more reliable expectation of a model's accuracy than the minimum alone.

**Final Displacement Error (FDE).** FDE is computed for each prediction $j$ as $\mathbf{b}_j$, the distance between the final positions of the ground truth and the prediction (Eq. 2) [1]. It is aggregated across the $k$ predictions in the same ways as ADE for better reliability.

$$\mathrm{ADE}\Big(\mathbf{Y}_a, \widehat{\mathbf{Y}}_a\Big) = \big[\Omega(\mathbf{a}), \Theta(\mathbf{a}), \mathrm{O}(\mathbf{a})\big] \tag{1}$$
$$s.t. \ \mathbf{a}_j = \frac{1}{t_f} \sum_{t=0}^{t_f-1} \Big\|\mathbf{Y}_{a,0,t} - \widehat{\mathbf{Y}}_{a,j,t}\Big\|, \ 0 \leq j < k$$

$$\mathrm{FDE}\Big(\mathbf{Y}_a, \widehat{\mathbf{Y}}_a\Big) = \big[\Omega(\mathbf{b}), \Theta(\mathbf{b}), \mathrm{O}(\mathbf{b})\big] \tag{2}$$
$$s.t. \ \mathbf{b}_j = \Big\|\mathbf{Y}_{a,0,t_f-1} - \widehat{\mathbf{Y}}_{a,j,t_f-1}\Big\|, \ 0 \leq j < k$$

## 4.3 Realism Metrics: Motion and Interaction Statistics

Realism metrics are used to describe the movement and interactions within the ground truth and the predictions separately. These metrics can then be used to uncover more nuanced differences between the ground truth and predictions. While they cannot ensure that predictions are accurate,

they can ensure that predictions are realistic in their movement and plausible. Every realism metric is computed in the same way for both the ground truth and predictions, so $\mathbf{Y}$ is interchangeable with $\widehat{\mathbf{Y}}$ and $\mathbf{V}$ with $\widehat{\mathbf{V}}$. For generality, we consider the ground truth as a unimodal model with $k = 1$, but we refer to it as having $k$ paths instead of predictions.

The following motion statistics are used to describe the movement of agent $a$ in either the ground truth or averaged across the $k$ predictions. They have been used to evaluate crowd simulations in computer graphics research [27], but have not yet been used to evaluate predictive models in computer vision.

**Path Length.** The average path length (m) for an agent $a$ is computed by first finding the length of each path $j$ and then averaging the values across all $k$ paths (Eq. 3).

**Speed.** In order to report the speed (m/s), the magnitudes $\mathbf{S} \in \mathbb{R}^{k \times (t_f - 1)}$ of velocities in $\mathbf{V}_a$ are first computed for each agent $a$. Next, two values are reported for speed: the mean speed averaged across $k$ paths and the maximum speed averaged across $k$ paths. For each path $j$ of agent $a$, the mean and maximum speed are computed across $t_f - 1$ frames (Eq. 4).

**Acceleration Magnitude.** Similar to speed, we first compute the magnitudes $\mathbf{A} \in \mathbb{R}^{k \times (t_f - 2)}$ of the difference between every pair of consecutive velocities in $\mathbf{V}_a$ for each agent $a$. The acceleration magnitude (m/s$^2$) $\mathrm{A}(\mathbf{V}_a)$ is then reported in the same way as speed: the mean acceleration magnitude averaged across $k$ paths and the maximum magnitude averaged across $k$ paths (Eq. 5).

$$\mathrm{L}(\mathbf{Y}_a) = \left[ \frac{1}{k} \sum_{j=0}^{k-1} \sum_{t=0}^{t_f-2} \left\| \mathbf{Y}_{a,j,t+1} - \mathbf{Y}_{a,j,t} \right\| \right] \tag{3}$$

$$\mathrm{S}(\mathbf{V}_a) = \left[ \frac{1}{k} \sum_{j=0}^{k-1} \Theta(\mathbf{S}_j) \, , \, \frac{1}{k} \sum_{j=0}^{k-1} \mathrm{O}(\mathbf{S}_j) \right] \tag{4}$$

$$s.t. \ \mathbf{S}_{j,t} = \left\| \mathbf{V}_{a,j,t} \right\|, \, 0 \le t < t_f - 1$$

$$\mathrm{A}(\mathbf{V}_a) = \left[ \frac{1}{k} \sum_{j=0}^{k-1} \Theta(\mathbf{A}_j) \, , \, \frac{1}{k} \sum_{j=0}^{k-1} \mathrm{O}(\mathbf{A}_j) \right] \tag{5}$$

$$s.t. \ \mathbf{A}_{j,t} = \left\| (\mathbf{V}_{a,j,t+1} - \mathbf{V}_{a,j,t})/\Delta t \right\|, \, 0 \le t < t_f - 2$$

Traditional measures of collision are unsuitable for multimodal models in which an agent $a$ may be colliding with agent $b$ when it takes the direction of path $j$, but not when it takes the direction of path $j + 1$. We therefore propose multiverse metrics such as Agent Collision-Free Likelihood (ACFL) and Environment Collision-Free Likelihood (ECFL) to measure the A2A and A2E interactions of multimodal models respectively.

**Agent Collision-Free Likelihood (ACFL).** In order to assess the quality of A2A interaction under the $k^n$ possible futures for $n$ agents, we propose ACFL, which computes the probability that agent $a$ has a path that is free of collision in all of the $k^{(n-1)}$ possible futures with other agents (Eq. 6). The indicator function $\mathbf{1}_{\mathbb{R}>0}$ returns 1 when the distance between agents $a$ and $b$ is greater than $r$ meters at time $t$, and 0 otherwise. This means that if their centers of mass are within $r$ meters of each other, they are considered to be colliding. For analysis, $r$ has been set to 0.3 meters ($\sim$1 foot).

**Environment Collision-Free Likelihood (ECFL).** ECFL complements ACFL in that it measures the quality of A2E interaction under the $k$ possible futures that agent $a$ can interact with the environment (Eq. 7). Namely, it reports the probability that agent $a$ has a path that is free of collision with the environment. The environment is represented by a binary matrix $\mathbf{E}$, in which each cell corresponds to a square space and is equal to 1 if that space is navigable and 0 otherwise. $\mathbf{E}[0,0]$ is aligned with the origin of the position data $\mathbf{Y}$, but $\mathbf{E}$ has a scale of $1/s$ meters per unit as opposed to 1 meter per unit like $\mathbf{Y}$. This means that the position $[x, y] = \mathbf{Y}_{a,j,t}$ of agent $a$ taking path $j$ at time $t$ maps to $\mathbf{E}\big[\lfloor s \cdot y \rfloor, \lfloor s \cdot x \rfloor\big]$. For analysis, $s$ has been set to 2 based on the dataset. When agent $a$'s center of mass is intersecting a non-navigable region of the environment like a wall, the agent is considered to be colliding with the environment.

$$\text{ACFL}(\mathbf{Y}, a) = \left[ \frac{1}{k} \sum_{j=0}^{k-1} \prod_{b=0}^{n-1} \prod_{i=0}^{k-1} \prod_{t=0}^{t_f-1} \mathbf{1}_{\mathbb{R}>0} \Big( \big\| \mathbf{Y}_{a,j,t} - \mathbf{Y}_{b,i,t} \big\| - r \Big) \right] \ s.t. \ a \neq b \qquad (6)$$

$$\text{ECFL}(\mathbf{Y}_a, \mathbf{E}) = \left[ \frac{1}{k} \sum_{j=1}^{k} \prod_{t=0}^{t_f-1} \mathbf{E} \Big[ \ \big\lfloor s \cdot \mathbf{Y}_{a,j,t,1} \big\rfloor, \big\lfloor s \cdot \mathbf{Y}_{a,j,t,0} \big\rfloor \ \Big] \ \right] \qquad (7)$$

$$\text{MVE}(\mathbf{Y}_a) = - \sum_{p \in \mathbf{p}} p \cdot \log_2(p) \ \ s.t. \ \ \mathbf{p} = \Xi\big( \mathbf{D}, 20 \big) \ , \qquad (8)$$

$$\mathbf{D}_j = \frac{1}{t_f - 1} \left( \sum_{t=1}^{t_f-1} \mathbf{Y}_{a,j,t} \right) - \mathbf{Y}_{a,j,0} \ , \ 0 \leq j < k$$

### 4.4   Decidability Metric: Certainty in Movement Direction

Decidability is a measure of a model's uncertainty in the movement direction of agents, and it is not strictly opposite between unimodal and multimodal models. If a multimodal model has low enough uncertainty in an agent's direction of movement, we consider it to be decidable.

**Multiverse Entropy (MVE).** We compute MVE to measure the decidability for agent $a$. We first transform each path $j$ into an average direction vector $\mathbf{D}_j \in \mathbb{R}^2$ as the vector from the initial position $\mathbf{Y}_{a,j,0}$ to the average position of the $t_f - 1$ subsequent points (Eq. 8). The average direction vectors $\mathbf{D}$ are then transformed into a probability distribution $\mathbf{p} \in \mathbb{R}^b$ over a vector of $b$-many equiangular bins (Fig. 3). Finally, the entropy of $\mathbf{p}$ is reported as MVE. High values of ACFL and ECFL are contingent on low MVE (high decidability), because high certainty in the direction that an agent will travel along will cause fewer potential collisions with other agents (ACFL) and the environment (ECFL). For experimental purposes, $b$ has been set to $k$, so that MVE is maximized when every prediction is in a different direction.

### 4.5   Comparing Realism Metrics

In order to compare realism metrics between the ground truth and predictions for an agent $a$, we first compute a feature vector for the ground truth $\mathbf{F}_a = \big\langle \text{L}(\mathbf{Y}_{a,0}), \text{S}(\mathbf{V}_a), \text{A}(\mathbf{V}_a), \text{ACFL}(\mathbf{Y}, a),$ $\text{ECFL}(\mathbf{Y}_a, \mathbf{E}) \big\rangle$, where $\langle \cdot, \cdot \rangle$ denotes vector concatenation. The same vector concatenation is used to compute the feature vector $\widehat{\mathbf{F}}_{a,j} \in \mathbb{R}^7$ for each prediction $j$. Equation 9 returns the percent differences $\widehat{\mathbf{C}}_a \in \mathbb{R}^k$ between the feature vectors of each prediction $j$ and the ground truth of agent $a$.

$$\widehat{\mathbf{C}}_{a,j} = \frac{100}{7} \sum_{f=0}^{6} \frac{\Big| \widehat{\mathbf{F}}_{a,j,f} - \mathbf{F}_{a,0,f} \Big|}{\mathbf{F}_{a,0,f}} \ \ s.t. \ \ \mathbf{F}_{a,0,f} > 0 \ , \ 0 \leq j < k \qquad (9)$$

## 5   Results

In order to understand the limits of not only the SOTA but also the models that paved the way towards the SOTA, we evaluate three critical multimodal models that are capable of either short-term or long-term trajectory prediction and provide a large coverage over the performance of prior models (Fig. 1.b). In particular, we have selected (1) Social GAN (SGAN) [7], one of the earliest models; (2) Trajectron++ (T++) [24], a SOTA model for short-term trajectory prediction; and (3) PECNet (PECN) [18], a SOTA model for long-term trajectory prediction.

**Training Protocol.** Each of the three models was trained on 3 combinations from the **A2X** Dataset: A2A interaction, A2E interaction, and both (Fig. 1.b), producing a total of 9 models. Each trained model was then evaluated on the testing sets of the 3 combinations (Fig. 1.c). The results of the evaluations on A2A and A2E are reported in Table 1, while the results on both A2A and A2E combined and corresponding visualizations are reported in the Supplementary Materials. According to the dataset, the following parameters have been set for the evaluation: $k = 20$, $t_p = 8$, $t_f = 12$,

and $\Delta t = 0.4$, meaning that each agent is receiving 3.2 seconds of input data and predicting 4.8 seconds into the future.

Each row of Table 1 reports the accuracy, realism, and decidability metrics of a model averaged across the agents of every testing scenario for a given dataset. The first 5 columns of realism metrics correspond to the dimensions of $\mathbf{F}$ and $\widehat{\mathbf{F}}$, the feature vectors used to compute the percent difference between the ground truth (GT) and predictions. The mean percent difference $\Theta(\widehat{\mathbf{C}}_a)$ of each agent $a$ is averaged across all agents and reported in the final column of the realism metrics. For all accuracy metrics, the realism percent difference, and the decidability metric, a lower value is favorable, while for the remaining realism metrics, a value closer to the ground truth is favorable.

| Test | Model | Train | Accuracy Metrics | | Realism Metrics | | | | | | Decidab. |
|---|---|---|---|---|---|---|---|---|---|---|---|
| | | | ADE ↓ min / mean / max | FDE ↓ min / mean / max | Length | Speed mean / max | Accel. mean / max | ACFL | ECFL | %Diff. ↓ | MVE ↓ |
| **Agent-to-Agent Interaction** | GT | N/A | 0.00 / 0.00 / 0.00 | 0.00 / 0.00 / 0.00 | 4.43 | 1.01 / 1.32 | 0.29 / 1.04 | 0.95 | 1.00 | 0 | 0.00 |
| | SGAN | A2A | **0.36** / 0.77 / 1.50 | **0.62** / 1.61 / 3.33 | **4.22** | **0.96** / 1.42 | 0.09 / **0.56** | 0.30 | **0.98** | **48** | 0.90 |
| | | A2E | 2.21 / 2.48 / 2.81 | 4.02 / 4.65 / 5.48 | 3.15 | 0.72 / 1.38 | **0.12** / 0.40 | 0.58 | 0.97 | 51 | **0.70** |
| | | Both | 0.37 / **0.74** / **1.35** | 0.65 / **1.55** / **2.97** | 4.13 | 0.94 / **1.32** | 0.06 / 0.33 | 0.33 | 0.98 | 51 | 0.84 |
| | PECN | A2A | **0.63** / **0.65** / **0.68** | **1.12** / **1.28** / **1.45** | 4.50 | 1.02 / 2.15 | 0.48 / **3.41** | 0.56 | 0.98 | **56** | **0.07** |
| | | A2E | 1.25 / 1.28 / 1.31 | 1.83 / 2.00 / 2.20 | **4.50** | **1.02** / 4.16 | 1.13 / 8.80 | **0.59** | **0.98** | 166 | 0.10 |
| | | Both | 0.73 / 0.76 / 0.79 | 1.44 / 1.59 / 1.74 | 4.78 | 1.08 / 2.61 | 0.49 / 4.57 | 0.57 | 0.98 | 85 | 0.10 |
| | T++ | A2A | **0.22** / 0.66 / 1.85 | **0.42** / 1.51 / 4.16 | **4.38** | **1.00** / 2.32 | **0.36** / 3.09 | 0.22 | **0.98** | 47 | **1.08** |
| | | A2E | 0.56 / 1.06 / 1.77 | 1.13 / 2.29 / **3.90** | 4.22 | 0.96 / **1.79** | 0.29 / **2.18** | 0.25 | 0.98 | 46 | 1.41 |
| | | Both | 0.23 / **0.64** / **1.76** | 0.43 / **1.48** / 4.02 | 4.35 | 0.99 / 2.27 | 0.35 / 2.96 | 0.22 | 0.98 | 47 | 1.13 |
| **Agent-to-Env. Interaction** | GT | N/A | 0.00 / 0.00 / 0.00 | 0.00 / 0.00 / 0.00 | 5.51 | 1.25 / 1.40 | 0.18 / 0.51 | 1.00 | 1.00 | 0 | 0.00 |
| | SGAN | A2A | 0.28 / 0.66 / 1.33 | 0.50 / 1.48 / 3.14 | **5.42** | **1.23** / 1.70 | 0.08 / **0.45** | 0.29 | 0.90 | **47** | 0.82 |
| | | A2E | **0.19** / **0.41** / **0.96** | **0.27** / **0.86** / **2.17** | 4.19 | 0.95 / **1.33** | **0.09** / 0.28 | **0.35** | **0.94** | 48 | **0.64** |
| | | Both | 0.19 / 0.56 / 1.25 | 0.32 / 1.28 / 3.02 | 5.03 | 1.14 / 1.57 | 0.08 / 0.40 | 0.32 | 0.92 | 49 | 0.65 |
| | PECN | A2A | 0.47 / 0.49 / 0.51 | 0.98 / 1.12 / 1.27 | **5.35** | **1.22** / 1.72 | **0.32** / 2.79 | **0.64** | 0.92 | **117** | **0.03** |
| | | A2E | **0.29** / **0.31** / **0.34** | **0.63** / **0.75** / **0.90** | 5.64 | 1.28 / 2.44 | 0.40 / 3.50 | 0.60 | **0.94** | 148 | 0.04 |
| | | Both | 0.32 / 0.34 / 0.37 | 0.70 / 0.81 / 0.92 | 5.64 | 1.28 / 2.29 | 0.34 / 3.41 | 0.60 | 0.93 | 157 | 0.06 |
| | T++ | A2A | 0.17 / 0.81 / 2.43 | 0.34 / 1.86 / 5.54 | 5.48 | 1.25 / 3.10 | 0.53 / 4.41 | 0.18 | 0.90 | 43 | 1.24 |
| | | A2E | **0.10** / **0.29** / **0.64** | **0.19** / **0.69** / **1.61** | **5.41** | **1.23** / **1.63** | 0.18 / **1.38** | **0.47** | **0.95** | **40** | **0.73** |
| | | Both | 0.12 / 0.37 / 1.11 | 0.23 / 0.87 / 2.55 | 5.41 | 1.23 / 2.00 | **0.27** / 2.04 | 0.42 | 0.93 | 40 | 0.76 |

Table 1: This table showcases the evaluation results of Social GAN (SGAN), PECNet (PECN), and Trajectron++ (T++) after training on either A2A, A2E, or both A2A and A2E and testing on A2A and A2E separately. For every metric in a testing set, the best value has been made bold for each model.

**Analysis.** As expected, we find that models trained on a single type of interaction perform poorly on test scenarios that feature the other type of interaction (Tab. 1). By training any of the three models on both types of interactions, we find that the accuracy of this model is either nearly the highest or the highest according to mean ADE/FDE compared to the same model trained on either A2A or A2E. For instance, T++$_{Both}$ trained on both types of interactions achieves the lowest mean ADE on A2A across all 9 trained models.

However, we cannot rely only on the accuracy of models to determine which is best, since anything short of perfect accuracy carries risk. The realism metrics allow us to better understand the model's performance in the context of its application. For example, we find that the maximum speed and acceleration for T++$_{Both}$ are significantly higher than the ground truth, which for an application in socially compliant robot navigation can discomfort or potentially harm surrounding humans [15]. In contrast, SGAN$_{Both}$ has lower average accuracy by a small margin, but it boasts higher realism by a large margin in terms of maximum speed, maximum acceleration magnitude, and ACFL. We attribute SGAN$_{Both}$'s higher ACFL to the tighter spread of its predictions than T++$_{Both}$ according to MVE. Ultimately, the choice of a model depends on the application, but without the joint consideration of the proposed accuracy and realism metrics, a practitioner may be led to choose an unsuitable model.

We have made 5 other notable observations from Table 1. (1) There are instances of models (highlighted in red) where relying on the optimistic lens of existing evaluations (i.e., minimum ADE and FDE) would lead to choosing models that are less accurate than others on average. (2) Models trained exclusively on A2E interactions tend to have lower likelihoods of A2A collision (higher ACFL) than models trained on A2A interactions alone or on both types of interactions, highlighting the important of A2E for improving robustness even in OOD scenarios such as A2A. (3) While this also holds true for the likelihood of A2E collision (ECFL) when testing on A2E, we find that ECFL is nearly perfect for A2A scenarios, indicating that A2A scenarios do not challenge models with A2E

interactions. (4) PECNet has the highest ACFL by an enormous margin owing to its MVE, which is low enough to consider PECNet as decidable and likely helps it in performing long-term trajectory prediction. Finally, (5) models trained on both types of interactions do not yet generalize to A2E better than models trained on A2E alone as some models have for A2A, meaning that there is still much room for improvement.

**Multimodal Model Collapse (MMC).** Accuracy metrics cannot be computed on never-before-seen data, because the ground truth is unknown. Consequently, it becomes impossible to find the predicted path with minimum error in accuracy and selecting an arbitrary prediction risks the maximum error. We therefore propose MMC, a baseline greedy method which can make use of the realism metrics to collapse the $k$ predictions of an undecidable multimodal model into a single well-informed prediction. In particular, we rely on the proposed comparison of realism metrics (Sec. 4.5), but instead of computing $\mathbf{F}_a$ from ground truth testing data $\mathbf{Y}_{a,0}$ for each agent $a$, we compute it as the average across *all* agents in the ground truth *training* data from the same environment. We then replace the $k$ predictions $\widehat{\mathbf{Y}}_a$ with the single prediction $j$ that minimizes the percent difference $\widehat{\mathbf{C}}_{a,j}$ for each agent $a$, which is the closest in realism to prior ground truth for the same type of scenario (Eq. 9). This, certainly, does not guarantee the optimal selection for a single agent. But it minimizes the overall error in selecting predictions for all agents. Table 2 shows the result of applying this technique to all 9 models. On average, we find that the ADE/FDE of the collapsed prediction is only ~15.76% worse than the mean ADE/FDE of the uncollapsed predictions, and ~31.63% better than the maximum ADE/FDE. Although the accuracy of the most realistic prediction is lower than the average accuracy over 20 predictions, its performance is consistently much better than the worst-case and it ultimately makes the undecidable model applicable to unseen data.

| Test | Model | Train | Accuracy Metrics | | Realism Metrics | | | | | | Decidab. |
|---|---|---|---|---|---|---|---|---|---|---|---|
| | | | ADE ↓ | FDE ↓ | Length | Speed | Accel. | ACFL | ECFL | %Diff. ↓ | MVE ↓ |
| | | | min = mean = max | min = mean = max | | mean / max | mean / max | | | | |
| | GT | N/A | 0.00 | 0.00 | 4.43 | 1.01 / 1.32 | 0.29 / 1.04 | 0.95 | 1.00 | 0 | 0.00 |
| Agent-to-Agent Interaction | SGAN | A2A | 0.91 | 1.99 | **4.28** | **0.97** / 1.20 | 0.16 / **0.41** | 0.69 | **0.99** | **37** | 0.00 |
| | | A2E | 2.57 | 4.97 | **3.75** | 0.85 / **1.32** | **0.20** / 0.37 | **0.79** | 0.97 | 40 | 0.00 |
| | | Both | **0.86** | **1.86** | 4.25 | 0.97 / 1.15 | 0.11 / 0.23 | 0.70 | 0.99 | 41 | 0.00 |
| | PECN | A2A | **0.65** | **1.27** | 4.44 | **1.01** / 1.56 | 0.33 / 1.79 | 0.66 | **0.98** | **56** | 0.00 |
| | | A2E | 1.28 | 2.03 | 4.33 | 0.98 / 3.23 | 1.02 / 6.37 | **0.68** | 0.98 | 166 | 0.00 |
| | | Both | 0.76 | 1.55 | 4.70 | 1.07 / 2.12 | 0.44 / 3.18 | 0.64 | 0.98 | 85 | 0.00 |
| | T++ | A2A | **0.81** | **1.83** | **4.51** | **1.03** / 1.31 | 0.44 / 0.98 | **0.66** | **0.99** | **26** | 0.00 |
| | | A2E | 1.05 | 2.27 | 4.53 | 1.03 / 1.32 | **0.42** / 0.97 | 0.63 | 0.98 | 30 | 0.00 |
| | | Both | 0.81 | 1.84 | 4.51 | 1.03 / 1.31 | 0.44 / **1.00** | 0.65 | 0.99 | 26 | 0.00 |
| | GT | N/A | 0.00 | 0.00 | 5.51 | 1.25 / 1.40 | 0.18 / 0.51 | 1.00 | 1.00 | 0 | 0.00 |
| Agent-to-Env. Interaction | SGAN | A2A | 0.76 | 1.84 | **5.00** | **1.14** / 1.44 | 0.15 / **0.33** | 0.63 | 0.96 | **38** | 0.00 |
| | | A2E | **0.69** | **1.60** | 4.73 | 1.08 / 1.30 | 0.13 / 0.23 | **0.68** | **0.98** | 40 | 0.00 |
| | | Both | 0.73 | 1.77 | 4.55 | 1.03 / **1.36** | **0.16** / 0.27 | 0.66 | 0.97 | 40 | 0.00 |
| | PECN | A2A | 0.49 | 1.11 | 5.39 | 1.22 / **1.45** | 0.25 / **1.10** | **0.69** | 0.93 | **117** | 0.00 |
| | | A2E | **0.30** | **0.71** | **5.54** | **1.26** / 1.71 | **0.31** / 1.41 | 0.62 | 0.93 | 148 | 0.00 |
| | | Both | 0.34 | 0.78 | 5.60 | 1.27 / 1.97 | 0.32 / 1.41 | 0.64 | **0.94** | 157 | 0.00 |
| | T++ | A2A | 0.90 | 2.06 | 4.99 | 1.13 / 1.48 | 0.57 / 1.27 | 0.46 | 0.97 | 31 | 0.00 |
| | | A2E | **0.34** | **0.86** | **5.36** | **1.22** / **1.44** | **0.29** / 0.85 | **0.61** | **0.98** | **24** | 0.00 |
| | | Both | 0.52 | 1.20 | 5.34 | 1.21 / 1.48 | 0.41 / **0.99** | 0.57 | 0.97 | 28 | 0.00 |

Table 2: This table reports the results of MMC on each of the 9 trained models. On average, MMC produces predictions that are consistently better than the worse case prediction prior to MMC. Only one value is reported for ADE and FDE, because the minimum, mean, and maximum are equal when $k = 1$. The MVE is always 0 when $k = 1$.

# 6 Conclusion

With the growing attention toward human trajectory prediction, it has become more important than ever to unify future research efforts in the right direction in terms of datasets and benchmark. In this work, we have brought to light the shortcomings of existing datasets, which hinder generalization, and existing evaluation metrics, which misrepresent model performance. By augmenting existing datasets with scenarios that feature substantial interactions between pedestrian agents and the environment, we have evidenced that models can generalize better. By proposing a comprehensive set of novel and existing evaluation metrics, we have not only proven the unreliability of existing evaluation metrics, but also highlighted the subtle factors that are essential for choosing the best trajectory prediction model for a particular application. Together, these contributions show that there is still room for much improvement even among the SOTA models.

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
