# OpenReview forum: "A2X: An Agent and Environment Interaction Benchmark for Multimodal Human Trajectory Prediction"
_NeurIPS.cc/2021/Track/Datasets_and_Benchmarks/Round1 — Submitted to NeurIPS 2021 Datasets and Benchmarks Track (Round 1)_

### Official Review · Reviewer_f6ww · 2021-06-29

**Rating:** 5
**Confidence:** 2
**Clarity:** The paper is well written and is easy…

**Strengths:**

+ The paper is well written and is easy to read.
+ Clear motivations and formulations
I think developing multiverse metrics to evaluate the multimodal models is quite challenging and is very important for the community of human trajectory prediction.

**Weaknesses:**

- The authors claim the new dataset facilitates better robustness and generalization, but it doesn't seem to be reflected in the experiments. And I think the authors should include ablative experiments that compare A2X with existing datasets.
- The effectiveness of MMC, As reported by the authors, is worse than the mean performance of *k* predictions. I wonder if you choose randomly from the *k* predictions, will the performance is the same as the mean performance, i.e. better than the proposed MMC.

**Additional Feedback:**

- The multiverse metrics treat all *k* predictions from the multimodal model with equal possibility. However, there exist many methods that predict results with different probabilities of confidence score. Therefore, I suggest the authors consider the case that *k* predictions with different weights.

**Correctness:**

The authors claim the new dataset enables better generalization. I think detailed experiments should be designed to validate this claim.

**Documentation:**

This paper provides details on data generation and organization. The data is available and the website is well maintained. But the documentation for dataset description, assumptions and evaluation should be added.

**Ethics:**

I do not see any ethical concerns in this paper.

**Relation To Prior Work:**

This paper is clearly discussed how this work differs from previous contributions.

**Summary And Contributions:**

This paper proposes a new A2X dataset to the generalization problem of existing human trajectory prediction datasets. Besides, the authors propose multiverse metrics to better evaluate current multimodal models. The results show the benefits of the dataset and metrics.

---

> ### Author Response · Authors · 2021-07-12
> **Response to Reviewer 3**
>
> We thank you for your review.
>
> Regarding the robustness and generalization of the new dataset, Table 1 evidences that when Social GAN (the seminal model) and Trajectron++ (a SOTA model) are trained on both existing datasets and the new dataset, average accuracy improves as a direct result of the new data when tested on the real-world test cases.
> Table 1 also highlights that models trained on existing datasets are unable to generalize to congested scenarios, which in practice carry a higher risk of injuries and casualties for involved pedestrians. In this critical case, the incorporation of the new dataset provides a drastic increase in accuracy.
> We are currently in the process of conducting the suggested ablative experiment, which will be incorporated into the Main Text.
>
> Regarding the effectiveness of MMC, Multimodal Model Collapse is introduced as a simple baseline to show that through the use of realism metrics, it is possible to significantly mitigate the worst-case performance, which is otherwise currently infeasible. As it stands, no SOTA multimodal model (e.g., Trajectron++ [1] or PECNet [2]) can guarantee that in a never-before-seen scenario, a particular predicted trajectory is not the worst-performing trajectory.
> Although a random prediction is more performant than MMC by ~16% (L322) in the average case, MMC is substantially more performant than the random prediction by ~32% (L323) in the worst case.
> Therefore, MMC is intended to serve as an avenue for future research in human trajectory prediction.
>
> Re the multiverse metrics for weighted predictions, we will note in the Main Text that for models with confidence scores for their predictions, the k predicted trajectories can be randomly resampled based on the softmax of their confidence scores.
>
> [1] Salzmann, Tim, et al. "Trajectron++: Dynamically-feasible trajectory forecasting with heterogeneous data." European Conference on Computer Vision. Springer International Publishing, 2020.
>
> [2] Mangalam, Karttikeya, et al. "It is not the journey but the destination: Endpoint conditioned trajectory prediction." European Conference on Computer Vision. Springer, Cham, 2020.

---

> > ### Comment · Reviewer_f6ww · 2021-07-15
> > **Response**
> >
> > I appreciate the detailed response from the authors.
> >
> > You say that SGAN and T++ models obtain performance improvement while trained on the new dataset. But for many metrics (e.g. MVE), I do not observe improvement. I think you can give more analyses on this phenomenon.
> >
> > Besides, regarding the multiverse metrics for weighted predictions, I do not mean randomly resampling. I mean when calculating the metrics like ACFL and ECFL, you can directly weigh the results with the predicted weights, not dividing them with *k*.
> >
> > Overall, I think this paper is valuable but many details need to be added and improved.

---

> > > ### Author Response · Authors · 2021-07-15
> > > **Response to Reviewer 3**
> > >
> > > We thank you for the clarification.
> > >
> > > T++ is overall the most accurate and most realistic model across both A2A and A2E, achieving the lowest min/mean/max ADE, min FDE, mean FDE on A2E, and percent difference in realism on both A2A and A2E (Table 1). It has also represented the existing SOTA in HTP for the three newest SOTA works: Y-Net [1], AgentFormer [2], and SGNet [3]. Accordingly, it carries a significant weight.
> > > We find that in terms of accuracy metrics (mean/max ADE and mean/max FDE), training on both A2A and A2E is definitively better than training on A2A alone (Table 1). This holds true not only for real-world test cases (A2A), but also for critical synthetic test cases (A2E) for which the improvement is even more pronounced (Table 1). In terms of realism metrics, the percent difference aggregates the differences between the prediction and ground truth in terms of the individual realism metrics. For T++, the percent difference indicates that training on the new synthetic dataset (A2E) alone is able to produce more realistic predictions than training on either A2A alone or both A2A and A2E together (Table 1).
> > > The MVE metric is an important indicator of undecidability, but training on A2A alone only marginally lowers the MVE from training on both A2A and A2E when testing on A2A (Table 1). On the contrary, when testing on A2E, the MVE of T++ trained on A2A alone is nearly double the MVE of T++ trained on the other datasets (Table 1).
> > > These findings reveal a comprehensive improvement in accuracy, realism, and decidability for T++ when training either jointly on existing datasets (A2A) and the new dataset (A2E) or exclusively on the new dataset (A2E).
> > >
> > > Regarding the multiverse metrics for weighted predictions, we will generalize the formulae in accordance with your suggestion. The original equal weighting was based on all recent SOTA multimodal models in HTP treating their predicted trajectories equally.
> > >
> > > [1] Mangalam, Karttikeya, et al. "From Goals, Waypoints & Paths To Long Term Human Trajectory Forecasting." arXiv preprint arXiv:2012.01526 (2020).
> > >
> > > [2] Yuan, Ye, et al. "AgentFormer: Agent-Aware Transformers for Socio-Temporal Multi-Agent Forecasting." arXiv preprint arXiv:2103.14023 (2021).
> > >
> > > [3] Wang, Chuhua, et al. "Stepwise Goal-Driven Networks for Trajectory Prediction." arXiv preprint arXiv:2103.14107 (2021).

---

> > > > ### Comment · Reviewer_f6ww · 2021-07-15
> > > > **Response**
> > > >
> > > > Thanks for your clarification. Now I finally understand how Table 1 shows the generalization of the proposed dataset. I think the authors should develop a better way to show your strength in the experiments, especially showing it in Table 1.

---

### Official Review · Reviewer_oQH7 · 2021-07-02
**The authors propose a simulated agent-to-environment dataset as well as three groups of metrics for human trajectory prediction. However, , the authors need to further explain what is the environment and the applications in real world.**

**Rating:** 4
**Confidence:** 4
**Correctness:** Yes
**Clarity:** Yes

**Strengths:**

They build a simulated A2X dataset for human trajectory prediction and evaluate the existing state-of-the-art methods.
They design three groups of metrics to evaluate the accuracy, realism and decidability of the methods.
The paper is well written and easy to follow.


**Weaknesses:**

My main concern focus on the understanding of the concept ‘environment’, which is very important for this work. However, the authors might fail to explain what is the environment or what does the environment include? The detailed and clear explanation is helpful to support the motivation and distinguish this work with the previous existing work. The authors always claim the existing dataset is with little to no agent-to-agent environment interaction. There is a question that does the environment include the building and cars in the Fig 1 (a) datasets? I think yes. Besides, there exists some work to use such surrounding semantic information to help the human trajectory prediction, such as [1] obeying physical constraints of the environment. Another important questions: what are the applications of the proposed dataset? How do we know the top-view structure map in real world? In the existing trajectory dataset, the trajectories are captured by the camera on the building or the drone. Besides, such trajectory dataset is more suitable to train RL to find the paths.

[1] Liang J, Jiang L, Niebles J C, et al. Peeking into the future: Predicting future person activities and locations in videos[C]//Proceedings of the IEEE/CVF Conference on Computer Vision and Pattern Recognition. 2019: 5725-5734.


**Additional Feedback:**

See the weakness.

**Documentation:**

Yes

**Relation To Prior Work:**

Yes

**Summary And Contributions:**

In this work, the authors mainly focus on human trajectory prediction. They propose a simulated agent-to-environment dataset as well as three groups of metrics. They also report and analyze the benchmark results.

---

> ### Author Response · Authors · 2021-07-12
> **Response to Reviewer 2**
>
> We thank you for your review.
>
> As explained using the taxonomy in Fig. 1(a) of the Main Text, A2E interactions are a critical facet of human navigational behavior that existing datasets do not represent. We propose a dataset that exemplifies these aspects and show empirical evidence that the incorporation of this data can improve the performance of SOTA models on real-world test cases.
>
> Regarding the environment's representation and dynamic obstacles (e.g., cars), our representation is the standard used in human trajectory prediction (HTP) and crowd flow prediction tasks [1,2,3].
> Specifically, the environment is a 2D binary matrix representation of the navigable spaces around static obstacles in a 2.5D physical environment, which resembles the floor plans of built environments. This implies that dynamic obstacles, such as vehicles, are not included, which is inconsequential to crowd flow within many built environments including most existing real-world datasets for human trajectory prediction [2].
>
> Regarding A2E interactions in existing datasets: while some datasets provide environment information, this does not imply that A2E interactions are occurring. In nearly all of these datasets, (1) pedestrians are moving in open spaces that do not necessitate pathfinding and (2) there are no bottlenecks in the environment to cause congestion, both of which are key types of A2E interaction.
>
> Regarding the applications of the proposed dataset: aside from the many potential applications of HTP that existing models cannot meet the demands of (e.g., architectural design, urban planning, transportation engineering, crowd management, and socially compliant robot navigation), the immediate application of the proposed dataset is in improving the accuracy, realism, and decidability of HTP models in real-world test cases. We empirically evidence the ability for the proposed dataset to do so in Table 1 of the main text.
>
> Re the acquisition of environment information, the top-down views of environments that suit the standard representation are almost ubiquitously available for real-world built environments through their floor plans.
> Re the application of RL models, the SOTA in HTP for existing datasets is currently dominated by models in computer vision literature (e.g., Trajectron++ [1], PECNet [4], and Y-Net [5]).
>
> [1] Salzmann, Tim, et al. "Trajectron++: Dynamically-feasible trajectory forecasting with heterogeneous data." European Conference on Computer Vision. Springer International Publishing, 2020.
>
> [2] Pellegrini, Stefano, et al. "You'll never walk alone: Modeling social behavior for multi-target tracking." 2009 IEEE 12th International Conference on Computer Vision. IEEE, 2009.
>
> [3] Sohn, Samuel S., et al. "Laying the foundations of deep long-term crowd flow prediction." European Conference on Computer Vision. Springer, Cham, 2020.
>
> [4] Mangalam, Karttikeya, et al. "It is not the journey but the destination: Endpoint conditioned trajectory prediction." European Conference on Computer Vision. Springer, Cham, 2020.
>
> [5] Mangalam, Karttikeya, et al. "From Goals, Waypoints & Paths To Long Term Human Trajectory Forecasting." arXiv preprint arXiv:2012.01526 (2020).

---

> > ### Comment · Reviewer_oQH7 · 2021-07-15
> > **Response**
> >
> > Thanks for the detailed authors' feedback. I have carefully read them, as well as other reviews. They address part of my questions. The remaining one is the acquisition of environment information, how to merge the top-down view and the human observed trajectories. It means when a person A walks in the building, how we know A' location on the top-down view during the inference? If we don't know or it's hard to get such information, the applications of the proposed dataset might be limited. The socially compliant robot navigation might also need the robot's location in the top-down view map. Overall, thanks very much for your response.

---

> > > ### Author Response · Authors · 2021-07-15
> > > **Response to Reviewer 2**
> > >
> > > We thank you for the clarification.
> > >
> > > In order to best illustrate this, we consider two potential applications of HTP: socially compliant robot navigation (SCRN) and architectural design. A perfect HTP model is essentially the best substitute for a real human's navigation.
> > > For SCRN, this can be used either to guide the robot's own navigation or to make the robot more cognizant of the humans around it. In either case, top-down environment information must be merged with trajectory information. This can be accomplished through the robot's calibration within the environment ahead of time and the use of visual SLAM algorithms in real-time [1]. SLAM can also be used to create the top-down environment information, which would be ready to merge with trajectory information as is.
> > > This methodology also extends to self-driving cars which make use of both GPS and SLAM algorithms in practice [2]. In this context, HTP is particularly relevant to urban environments, where cars are in close proximity to pedestrians.
> > > However, we are not restricted to using HTP models only in the real world. Design practitioners in architecture [3], urban planning [4], and transportation engineering [5] have historically used crowd simulations to study human interaction with built environments and to inform design decisions.
> > > Within a virtual environment, the merging of top-down environment information and agent trajectory information becomes trivial.
> > > Unlike prevalent handcrafted models [10], leveraging a perfect HTP model for crowd simulation would enable practitioners to gain the most accurate insights from their simulations, which are essential for evacuation planning, crowd management, and fire safety in the real-world [6, 7, 8, 9, 10].
> > >
> > > [1] Ibragimov, Ilmir Z., and Ilya M. Afanasyev. "Comparison of ROS-based visual SLAM methods in homogeneous indoor environment." 2017 14th Workshop on Positioning, Navigation and Communications (WPNC). IEEE, 2017.
> > >
> > > [2] Milz, Stefan, et al. "Visual slam for automated driving: Exploring the applications of deep learning." Proceedings of the IEEE Conference on Computer Vision and Pattern Recognition Workshops. 2018.
> > >
> > > [3] Gath-Morad, Michal, et al. "cogarch: Simulating wayfinding by architecture in multilevel buildings." 11th Annu. Symp. Simulation for Architecture & Urban Design (SimAUD 2020). ACM, 2020.
> > >
> > > [4] Lin, Ming C., and Dinesh Manocha. "Virtual cityscapes: recent advances in crowd modeling and traffic simulation." Frontiers of Computer Science in China 4.3 (2010): 405-416.
> > >
> > > [5] Sewall, Jason, et al. "Continuum traffic simulation." Computer Graphics Forum. Vol. 29. No. 2. Oxford, UK: Blackwell Publishing Ltd, 2010.
> > >
> > > [6] Ulicny, Branislav, and Daniel Thalmann. "Crowd simulation for interactive virtual environments and VR training systems." Computer animation and simulation 2001. Springer, Vienna, 2001. 163-170.
> > >
> > > [7] Johansson, Anders, et al. "From crowd dynamics to crowd safety: a video-based analysis." Advances in Complex Systems 11.04 (2008): 497-527.
> > >
> > > [8] Bohannon, John. "Directing the herd: Crowds and the science of evacuation." Science 310.5746 (2005): 219-221.
> > >
> > > [9] Thompson, Peter A., and Eric W. Marchant. "A computer model for the evacuation of large building populations." Fire safety journal 24.2 (1995): 131-148.
> > >
> > > [10] Helbing, Dirk, Illés Farkas, and Tamas Vicsek. "Simulating dynamical features of escape panic." Nature 407.6803 (2000): 487-490.

---

> > > > ### Comment · Reviewer_oQH7 · 2021-07-15
> > > > **Response-2**
> > > >
> > > > Thanks for your detailed feedback. Now, I understand the applications, in socially compliant robot navigation. Maybe the quite clear environment without any furniture in the rooms makes me have an unrealistic feeling, resulting in concerning the applications and the gaps between the real world and your simulated data. Besides, as shown in Fig. 3, the simulated top-down views are random and not so regular, far from the real scenes. This also makes me have concerns.

---

### Official Review · Reviewer_poXU · 2021-07-03
**A new agent-to-environment dataset**

**Rating:** 5
**Confidence:** 3
**Correctness:** yes, the metrics are well designed

**Strengths:**

A new agent-to-environment simulated dataset
Comprehensive metrics including accuracy, realism, and decidability metric

**Weaknesses:**

While the results are good, the dataset is simulated and lack real-world evidence

**Additional Feedback:**

no

**Clarity:**

yes, but the figures are sometimes hard to understand, e.g. Fig 1 is too complicated

**Documentation:**

yes the dataset has detailed information on the website https://mubbasir.github.io/HTP-benchmark

**Ethics:**

no, the dataset is simulated

**Relation To Prior Work:**

yes

**Summary And Contributions:**

The paper proposed a new simulated dataset that addressing the lack of agent-to-environment interaction in existing datasets. The paper also proposes a comprehensive set of accuracy, realism, and decidability metrics. Form the datasets and metrics, the author found evidences that the new dataset facilitates better robustness and generalization, that current metrics can be misleading, and that there are still remaining challenges to modeling human trajectories.

---

> ### Author Response · Authors · 2021-07-12
> **Response to Reviewer 1**
>
> We thank you for your review.
>
> Regarding the simulation of data and lack of real-world evidence, the collection of real data for high-density crowd interactions in large, complex environments at the scale needed for training machine learning models is simply infeasible.
> The particular method of simulation that we use has high ecological validity in our dataset's environments and has been adopted in numerous other disciplines (e.g., architecture, urban planning, and transportation engineering) as the preferred model of human navigation.
> Table 1 in the Main Text further evidences the realism of simulations, since synthetic data improves the average accuracy of Social GAN (the seminal model) and Trajectron++ (a SOTA model) on real-world test cases.
> The successful use of synthetic data (i.e., Domain Randomization) has been widely evidenced for other machine learning tasks in computer vision, reinforcement learning, and robotics, and serves as a precedent for our work. Notable examples include, Facebook's AI Habitat simulating movement in a photorealistic house, OpenAI simulating a Rubik's Cube for a robotic hand to solve, and NVIDIA simulating vehicular traffic for car detection.

---

### Decision · Program_Chairs · 2021-07-26

**Decision:**

Reject

**Comment:**

This paper focuses mainly on human trajectory prediction. They propose a simulated agent-to-environment dataset as well as three groups of metrics. They also report and analyze the benchmark results.
Scores assigned by the reviewers to this paper point towards rejection.
It seems like there are some misunderstandings of this paper, the authors tried to clarify them in the rebuttal. However, I would recommend the authors rewrite the paper to submit it to a future venue with a clearer structure and address the reviewers' concerns and misunderstandings.